# Maternal carriage of *Prevotella* during pregnancy associates with protection against food allergy in the offspring

Peter J. Vuillermin [1,2,3 ✉], Martin O'Hely[1,3], Fiona Collier[1,2,3], Katrina J. Allen[3,4,5], Mimi L.K. Tang[3,4,5], Leonard C. Harrison[5,6], John B. Carlin[3,4,5], Richard Saffery [3,5], Sarath Ranganathan[3,4,5], Peter D. Sly [3,7], Lawrence Gray[1,2,3], John Molloy [1,2,3], Angela Pezic[3], Michael Conlon[8], David Topping[8], Karen Nelson[9], Charles R. Mackay [10], Laurence Macia[11], Jennifer Koplin[3,5], Samantha L. Dawson[1,3], Margarita Moreno-Betancur[3,5] & Anne-Louise Ponsonby[3,5], the J. Craig Venter Institute*, the BIS Investigator Group*

In mice, the maternal microbiome influences fetal immune development and postnatal allergic outcomes. Westernized populations have high rates of allergic disease and low rates of gastrointestinal carriage of *Prevotella*, a commensal bacterial genus that produces short chain fatty acids and endotoxins, each of which may promote the development of fetal immune tolerance. In this study, we use a prebirth cohort ($n = 1064$ mothers) to conduct a nested case-cohort study comparing 58 mothers of babies with clinically proven food IgE mediated food allergy with 258 randomly selected mothers. Analysis of the V4 region of the 16S rRNA gene in fecal samples shows maternal carriage of *Prevotella copri* during pregnancy strongly predicts the absence of food allergy in the offspring. This association was confirmed using targeted qPCR and was independent of infant carriage of *P. copri*. Larger household size, which is a well-established protective factor for allergic disease, strongly predicts maternal carriage of *P. copri*.

[1] Deakin University, School of Medicine, Geelong, Victoria 3220, Australia. [2] Barwon Health, Geelong, Victoria 3220, Australia. [3] The Murdoch Children's Research Institute, Parkville, Victoria 3052, Australia. [4] The Royal Children's Hospital, Parkville, Victoria 3052, Australia. [5] The University of Melbourne, Parkville, Victoria 3052, Australia. [6] Walter and Eliza Hall Institute, Parkville, Victoria 3052, Australia. [7] University of Queensland, South Brisbane, Queensland 4101, Australia. [8] Commonwealth Science and Industrial Research Organisation, Adelaide, South Australia 5000, Australia. [9] J. Craig Venter Institute, Rockville, Maryland 20850, United States of America. [10] Monash University, Clayton, Victoria 3800, Australia. [11] University of Sydney, Charles Perkins Centre, Sydney, New South Wales 2006, Australia. *Lists of authors and their affiliations appear at the end of the paper. ✉email: peter.vuillermin@deakin.edu.au

I t has been 30 years since David Strachan first reported the association between larger household size and decreased allergic disease, leading to the so called hygiene hypothesis[1]. The ensuing decades have provided compelling evidence that diminished microbial exposure during early life is a risk factor for allergic disease, and that this association is likely to be mediated by changes in the human microbiome. More recently, it has been proposed that losses of specific bacterial species from our ancestral microbiota may be relevant to the increase in immune related diseases[2]; and that depletion in the compositional and functional diversity of the gut microbiome may be in part linked to a decrease in dietary fiber intake[3,4]. The maternal gut microbiome during pregnancy plays an important role in stimulating fetal immune development[5–8]; and cord blood immune phenotype at birth is associated with subsequent allergy[9,10]. To date, however, no human study has employed culture-independent techniques to investigate the relationship between maternal gut microbiota during pregnancy and subsequent allergic disease among offspring.

The commensal genus *Prevotella* is substantially less abundant in westernized populations than in traditional, non-industrial communities[11,12]. *Prevotella* are gram negative anaerobes that ferment dietary fiber to produce the short chain fatty acid (SCFA) acetate[13]. SCFAs have potent anti-inflammatory effects, in part by promoting the development of IL-10 producing regulatory T cells[14]. In pregnant mice, acetate produced by gut microbiota can cross the placenta and attenuate postnatal allergic responses in the offspring[6]. *Prevotella* also metabolize dietary fiber and fat to produce succinate[15], which stimulates innate immune cell development, migration and function[16]. In addition, *Prevotella* produces endotoxins, which influence fetal immune development and allergic outcomes via Toll-like receptor 4- dependent pathways[7]. Thus, it is plausible that low maternal carriage of *Prevotella* during pregnancy may be causally related to dysregulated immune development and high rates of allergic disease among children in westernized populations.

The aim of this study was to investigate the relationship between early life exposures, including maternal diet and household size, maternal and infant gut microbiota, and the risk of allergic disease during infancy. We hypothesized that greater maternal intake of dietary fiber during pregnancy is associated with decreased allergic disease in the offspring and that this association is mediated by SCFA producing gut organisms, in particular *Prevotella*.

We show in an Australian prebirth cohort, that larger household size predicts maternal carriage of *P. copri* during pregnancy,

which in turn strongly associates with protection against food allergy in the offspring.

## Results

**Participants.** The baseline characteristics of the inception cohort, the random subcohort, and food allergy case group are shown in Supplementary Table 1. Allergic sensitization was determined by skin prick allergy testing (SPT) at 1 year of age. Infants who were sensitized to a food were invited to undergo an in-hospital food challenge. The presence of wheeze and eczema during the first year was determined by parent report. Atopic wheeze and atopic eczema during the first year each required co-existing allergic sensitization at 1 year. The most common food allergy was to egg, followed by peanut, cashew and cow's milk (Supplementary Table 2). The definition of complete data for each analysis varied with the case definition: as shown in Fig. 1, the proportion of mother-infant dyads with complete data for the food allergy case-cohort study, the atopic wheeze sub-cohort study and the atopic eczema subcohort study were 294 of 362 (81%), 273 of 321 (85%) and 281 of 321 (88%), respectively. Within the random subcohort with complete data, 18 of 254 (7%) infants developed food allergy, 17 of 273 (6%) developed atopic wheeze and 14 of 281 (5%) developed atopic eczema by one year of age. Disregarding sensitization status, in the random subcohort with complete data, 130 of 281 (46.2%) infants had had a wheezing episode and 58 of 262 (22.1%) experienced eczema by one year of age.

**Maternal fecal microbiota and offspring allergy.** Hierarchical clustering based on OTU relative abundance was similar between technical replicates sequenced by MiSeq and analyzed according to the described bioinformatic pathway (Supplementary Fig. 1, Supplementary Methods). As shown in Fig. 2 and Supplementary Table 3, evidence of differential abundance of OTUs between food allergy case and non-case mothers was clearly strongest in relation to OTU41 and OTU697. OTU41 comprised 79% of all OTUs identified as belonging to the genus *Prevotella*. The sequence of 253 base pairs characterizing OTU41 was 100% identical to base pairs 529–781 of the *P. copri* strain JCM 13464 16S rRNA gene (Accession No: AB649279). The sequence characterizing OTU697 differed from that of OTU41 by 8 base pairs (96.8% identity, i.e. only one base pair mismatch more than would cause it to be included in OTU41). OTU697 was only evident in samples in which OTU41 was identified and their relative abundances were well correlated (r = 0.73, p < 0.001, n = 26; Pearson's product moment correlation and linear regression Wald test; Supplementary Fig. 2), suggesting OTU697 represents

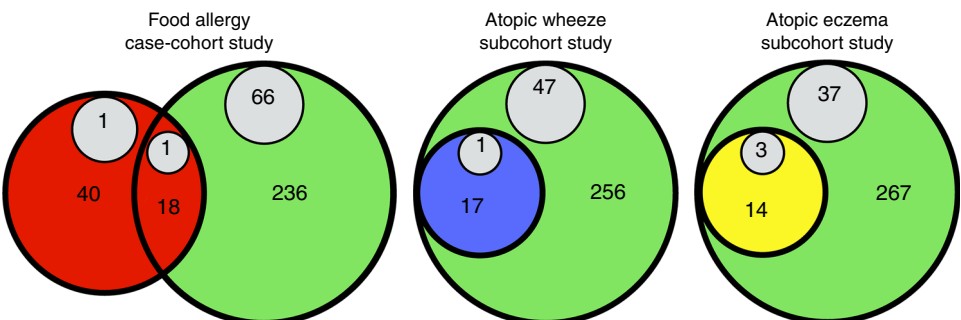

**Fig. 1 The assessment of three different allergic outcomes in infancy.** Food allergy was investigated using a case-cohort design, whereas atopic wheeze and atopic eczema were each investigated within the random subcohort only. The random subcohort is indicated by a large green circle (n = 321 mother-infant pairs). The case groups are indicated in red (food allergy), blue (atopic wheeze), and yellow (atopic eczema). The comparison groups are shown in green. Missing data, which varied with case definition, is indicated in gray.

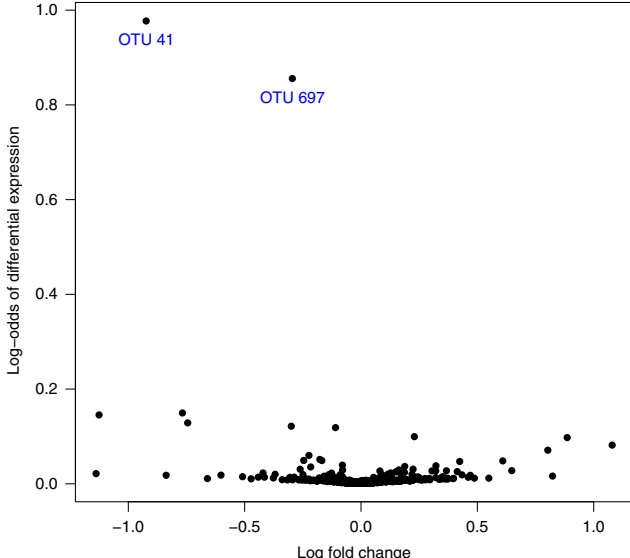

**Fig. 2 Volcano plot showing the magnitude (log-fold change) versus evidence (log-odds) of differential expression of all OTUs between food allergy case and non-case mothers.** A log-odds approaching 1.0 implies very strong evidence of differential expression. Evidence of differential expression was clearly strongest for OTU41 and OTU697.

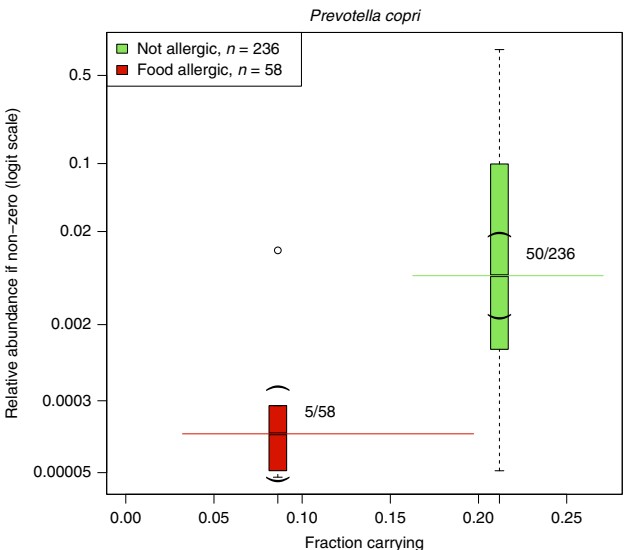

**Fig. 3 The fractional carriage and relative abundance of *P. copri* among mothers of infants with and without food allergy within the random subgroup.** The *x*-axis shows the fraction of mothers carrying *P. copri*, and the *y*-axis shows the relative abundance-when-present. Horizontal solid lines: 95% confidence intervals. Box plot elements: median (bar), first and third quartiles (box), the range of data within 1.5 times the inter-quartile range of the median (whiskers), and all data outside the whiskers shown explicitly; parentheses indicate the 95% CI of the median. Mothers of infants with food allergy ($n = 58$) compared with mothers of infants without food allergy within the random subgroup ($n = 236$).

either a minor variant of OTU41 or arose due to accumulated sequencing errors in reads that would otherwise have been assigned to OTU41.

As shown in Fig. 3, both the presence and abundance-when-present of *P. copri* (OTU41) was substantially greater among mothers of infants without food allergy compared to mothers of infants with food allergy ($p < 0.001$; limma test for differential

abundance via moderated *t*-test; $q = 0.003$; Benjamini–Hochberg correction). Indeed, there was only one case mother in whom *P. copri* > 0.03% was detected. Following adjustment for processing variables and potential confounding variables (household size, maternal dietary intake of polyunsaturated fatty acids, paternal history of allergic disease, pet ownership, and Caucasian ancestry; Supplementary Fig. 3) greater normalized abundance of *P. copri* continued to associate with strong evidence of protection against offspring food allergy ($p < 0.001$; limma test for differential abundance via moderated *t*-test). This evidence persisted following additional adjustment for peri- and post-natal variables (mode of delivery, perinatal antibiotics, breastfeeding, and age at introduction of solids) that may have been proxies for unmeasured antecedent confounding factors.

The presence of *P. copri* (OTU41) was positively associated with both the Shannon diversity index (difference of means –0.22 Shannon diversity units, Welch t-test 95%CI ($-0.39$, $-0.05$), $t = -2.54$, 100.3 d.f., $p = 0.01$) and the ratio of Bacteroidetes to Firmicutes (location difference –0.009, Wilcoxon rank-sum test with continuity correction 95%CI ($-0.030$, 0.001), $W = 4463$, $p = 0.07$). However, unlike *P. copri*, neither of these broader microbiome metrics was independently associated with food allergy in the offspring (Supplementary Fig. 4 and Supplementary Fig. 5). Beta diversity analysis did not reveal compelling differences in overall bacterial community structure between mothers of infants with and without allergy (Supplementary Fig. 6).

Targeted qPCR was then used to confirm the 16S findings. A doubling of *P. copri* expression in maternal stool was associated with an 8% decrease in the risk of food allergy in the children (adjusted risk ratio (aRR) 0.921, 95% CI 0.870 to 0.974, $p = 0.005$, $n = 276$; logarithmic regression Wald test). Substantial carriage of *P. copri*, which we defined as a qPCR expression estimate exceeding 1% (Supplementary Fig. 7), was associated with an 83% decrease in the risk of food allergy (aRR 0.165, 95%CI 0.039–0.693, $p = 0.02$, $n = 276$; logarithmic regression Wald test).

There was no evidence that the relationship between maternal carriage of *P. copri* and decreased food allergy was reflected in differences in the concentration of SCFAs in maternal feces (Supplementary Fig. 8).

Within the random subcohort, substantial expression of *P. copri* (qPCR) in maternal feces was associated with complete absence of offspring atopic wheeze (odds ratio via two-sided Fisher's exact test 0, 95%CI 0 to 0.936, $p = 0.03$, $n = 283$); and a trend toward decreased offspring atopic eczema (RR 0.280, 95% CI 0.038 to 2.074, $p = 0.2$, $n = 291$; logarithmic regression Wald test). However, disregarding sensitization status, there was no evidence that substantial expression of *P. copri* in maternal feces was associated with either wheeze (OR 0.88, 95%CI 0.48 to 1.62, $p = 0.8$, $n = 295$; two-sided Fisher's exact test) or eczema (RR 1.26, 95%CI 0.75 to 2.11, $p = 0.4$, $n = 272$; logarithmic regression Wald test) overall.

**Infant carriage of *P. copri*.** Maternal carriage of *P. copri* was positively correlated with infant carriage at 1, 6, and 12 months of age, as determined by 16S analysis at each timepoint (Supplementary Fig. 9). Of the three infant timepoints, the normalized abundance of *P. copri* at 6 months was most strongly associated with decreased food allergy at 1 year ($p < 0.001$; $q = 0.04$; $n = 312$; moderated *t*-test and Benjamini–Hochberg correction). However, the association between infant carriage of *P. copri* at 6 months and food allergy was substantially attenuated by adjustment for maternal carriage during pregnancy ($p = 0.85$; $q = 0.96$; $n = 312$; moderated *t*-test and Benjamini–Hochberg correction). By contrast, the association between maternal carriage of *P. copri* during

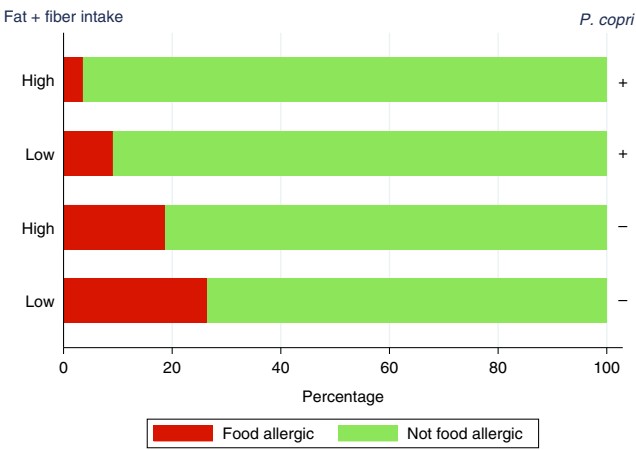

**Fig. 4 Maternal intake of fat and fiber, substantial carriage of *P. copri* and the offspring's risk of food allergy.** High: both fat and fiber intake are in the upper two quintiles (fiber greater than 22.4 g per day, fat greater than 77.3 g per day); low: one of fat and fiber intake is in the lower three quintiles.

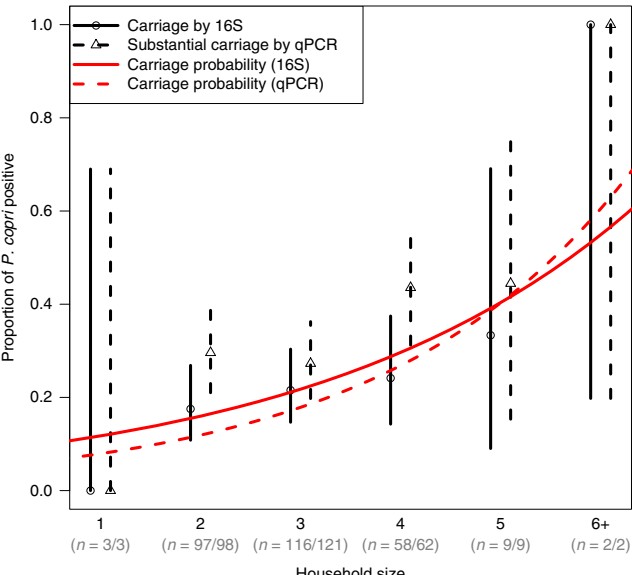

**Fig. 5 Household size and the proportion of mothers with *P. copri* detected in fecal samples collected during pregnancy.** Solid lines and circular symbols refer to *P. copri* carriage according to 16S sequencing, dashed lines and triangular symbols to substantial *P. copri* carriage according to qPCR analysis. Vertical lines indicate 95% CIs for the proportion and the red curves describe the predicted probability of carriage via logarithmic regression (risk ratio 1.36 per additional household member via 16S, 95%CI (1.13, 1.65), $p = 0.001$; RR 1.50 per additional member via qPCR, 95%CI (1.33, 1.70), $p < 0.001$; Wald test in logarithmic regression). The two largest households (containing six and seven members) are plotted as 6+ but were treated in the regression using their original values. Numbers of households in each size category are given for 16S/qPCR data available, respectively.

pregnancy and food allergy in the offspring persisted following adjustment for infant carriage at 6 months ($p < 0.001$; $q = 0.19$; $n = 277$; moderated $t$-test and Benjamini–Hochberg correction). In a formal mediation analysis using presence/absence of OTU41 in maternal and 6 month infant feces, the point estimate for the proportion of the total effect of maternal OTU41 carriage on food allergy mediated by infant carriage at six months was 12.1% (Supplementary Table 4).

**Maternal diet, *P. copri* and offspring allergy.** There was no evidence of an independent relationship between maternal fiber intake and either *P. copri* (OTU41) carriage or offspring allergy (Supplementary Fig. 10). *P. copri* metabolize dietary elements to produce both SCFAs and succinate[17], a carboxylic acid which stimulates innate immune cell development. As a diet high in both fat and fiber results in a synergistic increase in fecal succinate[15] we investigated the relationship between fat and fiber intake and offspring allergy. As shown in Fig. 4, infants of mothers with the highest two quintiles of both fiber and fat intake (fiber greater than 22.4 g per day, fat greater than 77.3 g per day) were at lower risk of food allergy compared to those with lower intake and this protective effect was greatest among women with substantial carriage of *P. copri*.

**Household size, antibiotics, and carriage of *P. copri*.** The hygiene hypothesis emerged from the observation that children from larger households were less likely to have allergic outcomes[1], an association that has been replicated in over 20 studies. As shown in Fig. 5, we found strong evidence that mothers in larger households were more likely to carry *P. copri*. There was a trend toward an association between household size and decreased offspring allergy mediated by maternal carriage of *P. copri* (Supplementary Table 5). Antibiotic use has also been associated with allergic outcomes in numerous studies, with mounting evidence that maternal exposure to antibiotics during pregnancy may be particularly important[18]. None of the 11 mothers in the random subcohort who reported being treated with antibiotics during the third trimester of pregnancy were *P. copri* (OTU41) carriers (Fisher exact $p = 0.13$, $n = 273$).

## Discussion
This is the first human study to use culture-independent techniques to investigate the relationship between the mother's gut

microbiota during pregnancy and allergic disease in the offspring. We found compelling evidence that maternal carriage of *P. copri* was associated with a decreased risk of clinically proven food allergy during infancy. The evidence of differential abundance was far stronger for two OTUs that mapped to *P. copri* than for those that mapped to other taxa, consistent with the microbial component of our a priori hypothesis. The magnitude of association was large, with evidence of dose-response, and was independent of a range of potential confounding factors. The protective association between maternal carriage of *P. copri* and decreased allergic disease in offspring was greatest among women with a diet high in fat and fiber. Increased household size and the absence of recent exposure to antibiotics were associated with increased maternal carriage of *P. copri*.

The strengths of this study include the unselected sampling frame, determination of food allergy status by clinical food challenge, rigorous assessment of potential confounding bias and confirmation of the 16S findings by qPCR analysis. Collection of both maternal and infant feces enabled us to confirm that the association between *P. copri* and decreased allergy primarily related to maternal carriage during pregnancy rather than offspring carriage during infancy. Limitations include the absence of metagenomic and metabolomic data from which to infer aspects of microbiome function, which might help elucidate the mechanisms by which maternal carriage of *P. copri* influences fetal immune development. Additionally, while the representative sequence for OTU41 exactly matched a 253 base pair span in *P. copri* strain JCM 13464, this does not preclude heterogeneity within the remainder of the bacterial genome, and the properties

of *P. copri* may vary between strains[19]. The use of a food frequency questionnaire may have also provided an inadequate estimate of diet, and in particular, intake of fiber. In addition, given changes in the composition of maternal microbiota between the first and third trimester[20], further studies are required to evaluate the relationship between maternal microbiota across the course of pregnancy and offspring outcomes.

The only previous human study to relate maternal fecal bacteria during pregnancy to allergy-related outcomes in the offspring compared fecal culture results obtained among 60 women during the third trimester of pregnancy with parent-reported wheeze and/or eczema during the first 6 months of infancy[21]. *Prevotella* is difficult to culture, being a strict anaerobe, and its carriage was not reported. However, a recent longitudinal analysis of 63 Swiss children found, using culture-independent techniques, underrepresentation of the genus *Prevotella* in fecal samples collected at 6 months, 13 months and 8 years from participants with IgE related allergic disease[22]. Consistent with this, a study of 83 South African children found a cross-sectional association between low relative abundance of *P. copri* and atopic dermatitis[23]. Here we have replicated an association between low *Prevotella* in postnatal life and allergy, but extended this finding by showing that it primarily reflects maternal carriage during pregnancy.

The association between maternal carriage of *P. copri* and decreased offspring allergy was not related to the concentration of SCFAs in maternal feces. SCFAs are rapidly consumed by large bowel microbiota and colonocytes, and accordingly, their concentrations drop substantially between the cecum and the anus[24]. Fecal measures may therefore be a poor proxy for the SCFA exposure of the fetus. Alternatively, the protective effect of *P. copri* may be mediated by other metabolites such as succinate. Consistent with this, inoculation of mice with *P. copri* results in increased cecal succinate without altering the concentration of SCFAs[17]. Although succinate stimulates inflammatory and migratory responses in innate immune cells[16], its role in fetal immune development has not been investigated. Other potential mechanisms include transplacental passage of *P. copri* epitopes bound to maternal IgG[5] or competitive inhibition of Toll-like receptor 4 signaling by the pentacylated form of endotoxin produced by *P. copri*. Finally, it is possible that *P. copri* may simply be a biomarker of other mechanisms underlying susceptibility to allergy.

The strong relationship between increased household size and maternal of carriage of *P. copri* is consistent with evidence that cohabiting individuals share microbiota, including Prevotellaceae[25]. Further studies are required to evaluate household transmission of *P. copri*. However, given the low rate of *Prevotella* carriage in modern communities, very large studies would be required to adequately evaluate whether the protective effect of larger household size on the risk of allergic disease is mediated by maternal carriage of *P. copri*.

Promoting maternal carriage of *P. copri* may have adverse effects. One study found a cross-sectional association between *P. copri* carriage and new-onset rheumatoid arthritis, and showed that *P. copri* exacerbates dextran-induced colitis in mice[26]. It has also been reported that, depending on fiber intake and strain variant, *P. copri* may either exacerbate or improve[17,27] insulin resistance. Therefore, future studies should consider potential pro-inflammatory and metabolic effects of *P. copri* strain variants as well as the interplay with diet.

Our findings have clear implications for public health, given the burden of allergic disease. The magnitude of effect was substantial and *P. copri* was undetected in ~80% of mothers. Consequently, if we assume causality, the estimated population-attributable risk of absence of maternal carriage of *P. copri* for food allergy is greater than 50%. Further studies are required to replicate the current findings in other populations, determine the underlying mechanisms and assess the potential of *P. copri* as a probiotic and/or biomarker. In the meantime, our findings support the importance of antibiotic stewardship during pregnancy as well as a diet that optimizes the health of the maternal gut microbiome.

## Methods

**Study design and population.** The Barwon Infant Study (BIS) is an Australian birth cohort study ($n = 1064$ mothers/1074 infants) assembled between 2010 and 2013 using an unselected antenatal sampling frame[28]. Mothers completed a food frequency questionnaire and provided a fecal sample in the third trimester of pregnancy. Infants were reviewed at 1, 3, 6, 9, and 12 months. A case-cohort design was used to compare infants with clinically proven food allergy ($n = 60$ mothers/61 infants) at 1 year with a random sample of the cohort ($n = 321$ mothers/324 infants). In the case-cohort analysis infants in the random subcohort with food allergy ($n = 19$) were classified as members of the case group. Within the random subcohort, we then compared mothers of children with and without (a) atopic wheeze, and (b) atopic eczema (Fig. 1). The study was approved by the ethics committee at Barwon Health and all mothers gave written informed consent before participating.

**Clinical outcome measures.** SPT was performed using Quintip® skin pricks to the following allergens: cow's milk, egg white, peanut, sesame, cashew, dust mite (*Dermatophagoides pteronyssinus* 1), cat, dog, rye grass, and *Alternaria tenius* (Stallergenes®). Sensitization to an allergen was defined as a SPT wheal size 2 mm or greater than the negative control, in the presence of a positive histamine control (≥2 mm). Infants with IgE-mediated food allergy were identified on the basis of a positive SPT at 12 months plus clinical history of a recent acute allergic reaction and/or formal in-hospital open food challenge using predetermined stopping criteria[29]. At each review, parents were asked about wheeze and eczema symptoms since the last review. Atopic wheeze was defined as the presence of parent-reported wheeze during the first year of life plus sensitization. Atopic eczema was defined as having eczema during the first year of life in accordance with the modified UK Working Party Diagnostic Criteria for Atopic Dermatitis for infants under 12 months of age plus sensitization.

**Diet and environmental exposures.** Maternal diet was recorded during the third trimester of pregnancy using the Dietary Questionnaire for Epidemiological Studies Version 2 (DQES2) to estimate daily macro and micronutrient intakes[30]. Maternal intake of antibiotics and microbial exposures during pregnancy were recorded by questionnaire at enrolment (28–32 weeks gestation).

**Fecal collection and processing.** Fecal samples were collected from pregnant women at 36 weeks gestation and infants at 1, 6, and 12 months and stored at −80 °C. DNA was extracted using the Qiagen PowerSoil® DNA Isolation Kit, Cat#12888-100 and transported to the J. Craig Venter Institute, Rockville, MD, USA. Universal primers for the V4 region of the 16S rRNA gene (Supplementary Table 6) were used to amplify a 292 bp product that was sequenced on the Illumina MiSeq platform. USEARCH[31] software was used to merge corresponding paired-end reads, filter (to remove merged reads with mismatches, too many or too few base pairs), cluster into OTUs at 97% identity, identify OTU representative sequences and remove chimeras. The mothur[32] software suite was used to assign representative sequences to taxa described in the SILVA v123 Nr99 taxonomic database. The final descriptions of OTUs present in each sample were composed in USEARCH. Samples with fewer than 2500 read pairs were excluded from further analysis. A small number of samples were processed with technical replicates for quality control checks but only the data for one sample per individual was used in the subsequent analyses. To confirm the main effect identified in the 16S analysis, primers and a Taqman probe specific to the *P. copri* 16S V4 region (Supplementary Table 6) were designed to identify *P. copri* carriers by real time PCR. Fecal samples were transported at −80 °C to the CSIRO laboratories, Adelaide, Australia, where SCFAs were quantified by capillary gas chromatography (GC; 5890 series II Hewlett Packard, Australia.)

**Statistical analysis.** The statistical software environment R (https://www.r-project.org), with the phyloseq[33] and limma packages[34], was used to manage the 16S data and conduct univariate and multivariable analyses. Raw OTU counts were normalized and weighted to account for variability in sequencing depth and normalized abundance was compared between cases and non-cases using moderated t-tests (eMethods). The voom function within limma leverages the data from technical replicates to improve statistical efficiency and performs better than competing methods (e.g. DESeq2[35]) for highly variable library sizes which are typical in 16S studies[34]. The log-fold change (LFC) reported by voom is based on abundances normalized using relative log expression with pseudo-counts added to avoid taking the log of 0 and including voom's precision weighting scheme. We refer to this metric as normalized abundance. The

Benjamini–Hochberg method was used to correct for multiple testing, its results termed $q$-values. We adjusted for the following processing variables: sequencing batch; duration of storage at −80 °C; and storage in home freezer prior to delivery. Potential confounding variables were determined from causal models represented in directed acyclic graphs (DAGs). Expression estimates from the qPCR analysis were treated both as log-transformed continuous measures (for which we added one half of the smallest positive estimate to all values so as to have a well-defined logarithm) and as a binary variable (substantial carriage, which we defined as a qPCR expression estimate exceeding 1%). In either case we estimated the risk ratio for offspring food allergy using binomial regression with a logarithmic link function and inverse probability weighting to account for the case-subcohort design. Median unbiased estimation and the midpoint-exact method for confidence interval estimation, as implemented in the R package epitools, were used to determine crude odds ratios for contingency table data on maternal diet, carriage of *P. copri* and offspring allergy. Change-in-estimate methods were used to evaluate confounding. Logarithmic regression was used to estimate the association between household size and maternal carriage of *P. copri* within the random subcohort. The R package survey[36] was used to perform these regressions. Mediation analysis was performed using the R package medflex[37], where the neWeight approach allowed us to incorporate inverse probability weighting. Any statistical tests with one- or two-sided alternatives were performed as two-sided.

**Reporting summary**. Further information on research design is available in the Nature Research Reporting Summary linked to this article.

## Data availability

Microbiota sequencing reads have been submitted to the Sequence Read Archive under accession number PRJNA576314 [https://www.ncbi.nlm.nih.gov/sra/?term=PRJNA576314]. Access to BIS data including all data used in this paper can be requested through the BIS Steering Committee by contacting the corresponding author. Requests to access cohort data are considered on scientific and ethical grounds and, if approved, provided under collaborative research agreements. Deidentified cohort data can be provided in Stata or CSV format. Additional project information, including cohort data description and access procedure, is available at the project's website https://www.barwoninfantstudy.org.au. Source data underlying Figs. 2–5 and Supplementary Figs. 2, 4–10 have been provided as a Source Data file.

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

## Acknowledgements

We would like to thank the study participants, as well as the entire BIS team, which includes interviewers, nurses, computer and laboratory technicians, clerical workers, research scientists, volunteers, managers, and receptionists. We also thank the obstetric and midwifery teams at Barwon Health and Saint John of God Hospital Geelong for their assistance in recruitment and collection of biological specimens. This study was funded by the National Health and Medical Research Council of Australia (1082307, 1147980), the Australian Food Allergy Foundation, The Murdoch Children's Research Institute, Barwon Health and Deakin University.

## Author contributions

P.V., A.L.P., K.J.A., M.L.T., S.R., J.B.C., and R.S., conceived of the study and designed the protocol. M.O.H., P.V., F.C., A.L.P., M.M., and A.P., conducted the analyses. P.V. wrote the original draft of the paper. A.L.P., M.O.H., F.C., K.J.A., M.L.T., L.C.H., J.B.C., R.S., S.R., P.D.S., J.M., M.M., A.P., M.C., D.T., K.N., C.M., L.M., L.G., J.K., and S.D. critically revised successive drafts of the paper and approved its final version.

## Competing interests

The findings described in this paper are the subject of a patent, licensed to Prevatex Pty Ltd, in which the following authors have a financial interest: P.V., M.O.H., F.C., M.T., S.R., A.L.P. The other authors declare no competing interests.

## Additional information

## the J. Craig Venter Institute

Sanjay Vashee[9], Manolito Torralba[9] & Andres Gomez[9]

## the BIS Investigator Group

Terrence Dwyer[3], David Burgner[3], Michael Forrester[1,2], Christos Symeonides[3] & Esther Bandala Sanchez[6]

