## [Peer Review File · Nature Communications]

Reviewers' comments:

Reviewer #1 (Remarks to the Author):

Thank you for allowing me to review this nice manuscript.

Here, Vuillermin and colleagues present the results a subgroup from their prospective mother-child cohort study (n=1064). They utilize maternal gut microbiota samples at week 36 of pregnancy along with childhood gut samples to evaluate the effects of these microbial compartments for allergy development in the children – they examine this in nested case-cohort study (n=362 mothers). The primary outcome measure was offspring food allergy status determined by skin prick testing and oral food challenge at 1 year of age. They also want to examine the early life exposures in relation to maternal gut microbiota. *A priori* they had the hypothesis that *Prevotella* might protect from allergy, perhaps through short chain fatty acids. They show that carriage of 2 OTUs suggested to be *Prevotella copri* in the maternal fecal samples seem to be protective from proven food allergy and atopic wheeze. The same appeared for the children's *P. copri*, but not as strong, and the authors conclude that maternal composition might be the most important.

The study is well performed, the prospective cohort setup is an advantage and the text is clearly written. The findings could be of high clinical importance in the future, and may add to our understanding of allergic disease inception, specifically the mother vs child carriage. The findings appear strong and consistent. The group of women whose children did not develop allergy had both a higher carriage and also if they were carriers, a higher abundance of *P. copri*.

The methodology for determining the microbial compositions from 16S rRNA sequencing of the V4 region and the lab and bioinformatics pipeline seem standard for the field. It is a strength that they verified *Prevotella* carriage by qPCR. Statistical methods are appropriate.

Given the style of Nature Communications – with methods section after the results, I would suggest a small paragraph added in the results section that describes a definition of the clinical phenotypes – age of diagnoses, and when the sensitization tests were performed. Furthermore both the atopic wheeze and the atopic dermatitis diagnoses require sensitization. How does the correlations look without this requirement? *P. copri* vs any wheeze, and *P. copri* vs eczema. My point is, whether the associations with wheeze and AD are just consequences of the strong allergy signal?

Maternal vs childhood *P. copri* carriage is an interesting part, as the authors find that maternal has a stronger association than the children's own carriage. The authors do analyses mutually adjusted for carriage; perhaps it could be worthwhile doing a real mediation analysis. How much of the maternal effect is through childhood carriage, and vice versa. Also it would be a nice addition to figure E9 to see how big a part of the childhood samples positive for *P. copri* had a positive maternal sample.

In the introduction they suggest that an immune programming may take place in pregnancy. In line with this, do the cohort have any data showing immunological support of a pregnancy rather than childhood effect of *P. copri*?

The authors open up an interesting point, when comparing *P. copri* with family size, and they perform a mediation analysis based on this. Is family size in itself associated with allergy in the BIS cohort?

How does the main study effect of *P. copri* vs. allergy look when adjusted for family size?

Figure 4 – I would suggest to use the term food allergy instead of FA in the legend.

Can the authors speculate whether carriage in itself or a high abundance is the most important for the protective effects?

Were any *Prevotella* OTUs identified, which did not associate with allergy? Perhaps because of low numbers. And could these still be *P. copri*?

Reviewer #2 (Remarks to the Author):

In this manuscript, the authors show that maternal carriage of *Prevotella copri* during the third trimester of pregnancy is associated with the lower risk of food allergy development in offspring by one year of age. Larger household size that has been suggested as a protective factor for allergic disease was also associated with maternal carriage of *P. copri*. This is a well-conducted study that employs microbiota analysis both by 16S rRNA sequencing and quantitative PCR, clinically proven

food allergy, and assessment of both maternal and infant fecal microbiota. The association of maternal carriage of *P. copri* with the protection of offspring towards food allergy is strong and novel. Analyses of microbiota and statistics are logically performed. The strengths and the limitations of the study are also discussed. My major concerns are, as the authors also discussed, the lack of the extension of this finding to different populations, the identification of the critical time frame of maternal *P. copri* during pregnancy, and providing the mechanisms by which maternal *P. copri* carriage during pregnancy may exert a protective effect towards food allergy. These would have increased the significance of the study. Nevertheless, I found the topic and the results are of an interest for the readers in the field. My specific comments are summarized below.

1. The authors focused on maternal microbiota during pregnancy, however, human milk has been suggested as a major determinant of infant gut microbiota (Pannaraj PS, JAMA Pediatr 2017). Also, solid food introduction alters infant gut microbiota (Subramanian S et al., Nature 2014 etc). It has been stated that the association of maternal *P. copri* with the protection of offspring against food allergy remained significant after adjustment for breastfeeding and age of solid food introduction, but these information should be included in the subject characteristics.
2. The definition of a diet “high in both fat and fiber” should be provided.
3. The household size and carriage of *P. copri* has been associated, but the association of the household size with the incidence of offspring food allergy ** within this cohort ** should be provided.

18 October 2019

Thank you for the opportunity to submit a revised manuscript. Our thanks also to the reviewers for recognising the importance of this study, as well as their questions and suggestions. Please find our responses below:

Reviewer #1 (Remarks to the Author):

Here, Vuillermin and colleagues present the results a subgroup from their prospective mother-child cohort study (n=1064). They utilize maternal gut microbiota samples at week 36 of pregnancy along with childhood gut samples to evaluate the effects of these microbial compartments for allergy development in the children – they examine this in nested case-cohort study (n=362 mothers). The primary outcome measure was offspring food allergy status determined by skin prick testing and oral food challenge at 1 year of age. They also want to examine the early life exposures in relation to maternal gut microbiota. A priori they had the hypothesis that Prevotella might protect from allergy, perhaps through short chain fatty acids. They show that carriage of 2 OTUs suggested to be Prevotella copri in the maternal fecal samples seem to be protective from proven food allergy and atopic wheeze. The same appeared for the children's P. copri, but not as strong, and the authors conclude that maternal composition might be the most important.

The study is well performed, the prospective cohort setup is an advantage and the text is clearly written. The findings could be of high clinical importance in the future, and may add to our understanding of allergic disease inception, specifically the mother vs child carriage. The findings appear strong and consistent. The group of women whose children did not develop allergy had both a higher carriage and also if they were carriers, a higher abundance of P.copri.

The methodology for determining the microbial compositions from 16S rRNA sequencing of the V4 region and the lab and bioinformatics pipeline seem standard for the field. It is a strength that they verified Prevotella carriage by qPCR. Statistical methods are appropriate. Given the style of Nature Communications – with methods section after the results, I would suggest a small paragraph added in the results section that describes a definition of the clinical phenotypes – age of diagnoses, and when the sensitization tests were performed.

Response 1.1: The following brief description of the clinical phenotypes has been added to the Results section:

“Allergic sensitization was determined by skin prick allergy testing (SPT) at 1 year of age. Infants who were sensitized to a food were invited to undergo an in-hospital food challenge. The presence of wheeze and eczema during the first year was determined by parent report. Atopic wheeze and atopic eczema during the first year each required co-existing allergic sensitization at 1 year.” Page 5, para 1.

Furthermore both the atopic wheeze and the atopic dermatitis diagnoses require sensitization. How does the correlations look without this requirement? P. copri vs any wheeze, and P. copri vs eczema. My point is, whether the associations with wheeze and AD are just consequences of the strong allergy signal?

Response 1.2: The following has been added to the Results section:

“However, disregarding sensitization status, there was no evidence that substantial expression of *P. copri* in maternal feces was associated with either wheeze (OR 0.88, 95%CI 0.48 to 1.62, p=0.8, N=295) or eczema (RR 1.26, 95%CI 0.75 to 2.11, p=0.4, N=272) overall.” Page

7, para 4.

Maternal vs childhood P. copri carriage is an interesting part, as the authors find that maternal has a stronger association than the children's own carriage. The authors do analyses mutually adjusted for carriage; perhaps it could be worthwhile doing a real mediation analysis. How much of the maternal effect is through childhood carriage, and vice versa.

Response 1.3: The following has been added to the Results section:

“In a formal mediation analysis using presence/absence of OTU 41 in maternal and 6 months infant feces, the point estimate for the proportion of the total effect of maternal OTU41 carriage on food allergy mediated by infant carriage at six months was 12.1% (eTable 4).”

Page 8, para 1.

On temporal grounds, it is not possible that relationship between infant carriage of *P. copri* and decreased allergy could be mediated by a preceding variable i.e. maternal carriage during pregnancy. We have therefore not included an estimate of the proportion of the total effect of infant carriage of OTU41 at 6 months that is mediated by maternal carriage during pregnancy.

Also it would be a nice addition to figure E9 to see how big a part of the childhood samples positive for P. copri had a positive maternal sample.

Response 1.4: Thank you for this suggestion. Figure E9 has been updated to indicate the proportions of infants carrying *P. copri* according to maternal carriage.

In the introduction they suggest that an immune programming may take place in pregnancy. In line with this, do the cohort have any data showing immunological support of a pregnancy rather than childhood effect of P. copri?

Response 1.5: We found evidence that maternal carriage of *P. copri* is associated a higher naïve regulatory T cells as a proportion of lymphocytes (nTreg/lymphocytes) in cord blood: In an analysis restricted to the random subcohort (the appropriate sampling frame to avoid collider bias due to over-representation of food allergy cases within the case-cohort sample), substantial carriage of *P. copri* was associated with a 0.20 log units increase in nTreg/lymphocytes (95% CI (0.04, 0.36); p=0.016). This evidence weakened with adjustment for exposure to labour: 0.15 log units of Treg/lymphocytes (95%CI (-0.006, 0.312); p=0.059).

In turn, an analysis among the case-cohort (the appropriate sampling frame for questions relating to food allergy) revealed evidence that an elevated proportion of nTreg/lymphocytes in cord blood was associated with reduced food allergy: odds ratio 0.21 per unit change in log(nTreg/lymphocytes), (95% CI (0.05, 0.83); p=0.03). This evidence was essentially unchanged following adjustment for exposure to labour: odds ratio 0.20 per unit change in log(treg ratio), (95%CI (0.05, 0.76); p=0.02).

Though intriguing, we do not think these findings are sufficiently convincing to warrant inclusion in the manuscript. Further studies are needed to identify the underlying basis of the strong association between maternal carriage of *P. copri* and decreased offspring allergy reported in this paper.

The authors open up an interesting point, when comparing P. copri with family size, and they perform a mediation analysis based on this. Is family size in itself associated with allergy in the BIS cohort?

Response 1.6: We observe an odds ratio of 0.88 (95%CI (0.63, 1.20), p=0.4) for food allergy per additional older sibling. The evidence of association was stronger when we relate 2 or more children less than 6 years of age in the household during pregnancy to the risk of food allergy: odds ratio of 0.24 (95% CI (0.03, 0.92), p=0.04) (Fisher exact test). Given that over 20 previous studies have found associations between increased household size and decreased allergic disease, we investigated the statistical power of our study to identify an analogous association. To do this we conducted a simulation using the magnitude of association between household size and eczema reported by Strachan (*BMJ* 1989), inflated to account for a slightly higher prevalence of food allergy in BIS, and with counts of older sibling categories as collected in BIS. The power to detect the trend reported by Strachan is about 20% using an alpha of 0.05 — confirming that this aspect of our analyses is underpowered. We have not included these findings in the manuscript but would be happy to do so if requested.

How does the main study effect of P. copri vs. allergy look when adjusted for family size?

Response 1.7: We now include household size among the adjustment variables and there is no material change to the reported results. Page 6, para 2.

Figure 4 – I would suggest to use the term food allergy instead of FA in the legend.

Response 1.8: Figure 4 has been updated with this change.

Can the authors speculate whether carriage in itself or a high abundance is the most important for the protective effects?

Response 1.9: Both the rate of carriage of *P. copri* and the abundance-when-present were lower among the mothers of infants with food allergy (Figure 3, reproduced below). If carriage was the most important factor, the boxes in would be aligned vertically and separated horizontally (green to the right of red). If abundance were most important then the boxes would be aligned horizontally and separated vertically (green directly above red). The boxes are clearly separated both vertically and horizontally, suggesting both carriage and abundance-when-present are relevant, but we unable to determine which is the more important factor.

Prevotella copri

Fig 3. The fractional carriage and relative abundance of *P. copri* among mothers of infants with food allergy (n=58) compared with mothers of infants without food allergy within the random subgroup (n=236). The X-axis shows the fraction of mothers carrying *P. copri*, and the Y-axis shows the relative abundance-when-present. Horizontal solid lines: 95% confidence intervals. Vertical dashed line: range of count values, with parentheses indicating the 95% CI of the median.

Were any Prevotella OTUs identified, which did not associate with allergy? Perhaps because of low numbers. And could these still be P. copri?

Response 1.10: OTU41 which we identify as *P. copri* represented 97% of reads in maternal samples classified as *Prevotella_9* in the SILVA database. Four other *Prevotella_9* OTUs were identified, whose representative sequences had either 8 or 9 mismatches with that of OTU41 (two examples each). The 8-mismatch OTUs represented 2% (OTU697, commented on in the text) and 1% (OTU677, protective like 41 and 697, p=0.007, q=0.39) of the *Prevotella_9* reads. The 9-mismatch OTUs were excluded from the limma/voom analysis by the variance threshold filter.

Another 11 OTUs belonging to other subsets of the SILVA *Prevotella* or Prevotellaceae (the latter not identified at the genus level) had little evidence for association with food allergy. Being outside the *Prevotella_9* group these would not be expected to be *P. copri*.

Reviewer #2 (Remarks to the Author):

In this manuscript, the authors show that maternal carriage of Prevotella copri during the third trimester of pregnancy is associated with the lower risk of food allergy development in offspring by one year of age. Larger household size that has been suggested as a protective factor for allergic disease was also associated with maternal carriage of P. copri. This is a well-conducted study that employs microbiota analysis both by 16S rRNA sequencing and quantitative PCR, clinically proven food allergy, and assessment of both maternal and infant fecal microbiota. The association of maternal carriage of P. copri with the protection of offspring towards food allergy is strong and novel. Analyses of microbiota and statistics are logically performed. The strengths and the limitations of the study are also discussed. My major concerns are, as the authors also discussed, the lack of the extension of this finding to different populations, the identification of the critical time frame of maternal P. copri during pregnancy, and providing the mechanisms by which maternal P. copri carriage during pregnancy may exert a protective effect towards food allergy. These would have increased the significance of the study. Nevertheless, I found the topic and the results are of an interest for the readers in the field. My specific comments are summarized below.

1. The authors focused on maternal microbiota during pregnancy, however, human milk has been suggested as a major determinant of infant gut microbiota (Pannaraj PS, JAMA Pediatr 2017). Also, solid food introduction alters infant gut microbiota (Subramanian S et al., Nature 2014 etc). It has been stated that the association of maternal P. copri with the protection of offspring against food allergy remained significant after adjustment for breastfeeding and age of solid food introduction, but these information should be included in the subject characteristics.

Response 2.1: These data have now been included in eTable 1.

2. The definition of a diet “high in both fat and fiber” should be provided.

Response 2.2: The following definition has been added to the Results section:

“As shown in Fig 4, infants of mothers with the highest two quintiles of both fiber and fat intake (fiber greater than 22.4 g/day, fat greater than 77.3 g/day) were at lower risk of food allergy compared to those with lower intake and this protective effect was greatest among women with substantial carriage of *P. copri*.” Page 9, para 1.

3. The household size and carriage of P. copri has been associated, but the association of the household size with the incidence of offspring food allergy within this cohort should be provided.

Response 2.3: Please see our response above to a similar comment regarding household size and offspring food allergy from Reviewer 1 (R1.6).

Thank you for considering our revised manuscript for publication in Nature Communications.

Best wishes,
Peter Vuillermin

REVIEWERS' COMMENTS:

Reviewer #1 (Remarks to the Author):

All my comments and concerns have been sufficiently addressed.

Thank you for allowing me to review the manuscript,

Jakob Stokholm

Reviewer #2 (Remarks to the Author):

The authors sufficiently answered to my questions.